# Acyclovir-Loaded Solid Lipid Nanoparticles: Optimization, Characterization and Evaluation of Its Pharmacokinetic Profile

**DOI:** 10.3390/nano10091785

**Published:** 2020-09-09

**Authors:** Haniza Hassan, Ramatu Omenesa Bello, Siti Khadijah Adam, Ekram Alias, Meor Mohd Redzuan Meor Mohd Affandi, Ahmad Fuad Shamsuddin, Rusliza Basir

**Affiliations:** 1Department of Human Anatomy, Faculty of Medicine and Health Sciences, University Putra Malaysia (UPM), Serdang 43400, Selangor, Malaysia; ramatu42@yahoo.com (R.O.B.); sk.adam@upm.edu.my (S.K.A.); rusliza@upm.edu.my (R.B.); 2Department of Pharmacology and Therapeutics, Faculty of Pharmaceutical Sciences, Ahmadu Bello University, Samaru Zaria 810271, Kaduna State, Nigeria; 3Department of Biochemistry, Faculty of Medicine, Universiti Kebangsaan Malaysia, UKM Medical Centre, Jalan Yaakob Latiff, Bandar Tun Razak 56000, Kuala Lumpur, Malaysia; ekram.alias@ppukm.ukm.edu.my; 4School of Pharmacy, Universiti Teknologi MARA (UiTM), Puncak Alam Campus, Bandar Puncak Alam 42300, Selangor, Malaysia; meor@uitm.edu.my; 5Faculty of Pharmacy and Health Sciences, Universiti Kuala Lumpur Royal College of Medicine Perak, Ipoh 30450, Perak, Malaysia; fuad.shamsuddin@unikl.edu.my

**Keywords:** acyclovir, bioavailability, oral delivery, solid lipid nanoparticles

## Abstract

Acyclovir is an antiviral drug used for the treatment of *herpes simplex* virus infection. Its oral bioavailability is low; therefore, frequent and high doses are prescribed for optimum therapeutic efficacy. Moreover, the current therapeutic regimen of acyclovir is associated with unwarranted adverse effects, hence prompting the need for a suitable drug carrier to overcome these limitations. This study aimed to develop solid lipid nanoparticles (SLNs) as acyclovir carriers and evaluate their in vivo pharmacokinetic parameters to prove the study hypothesis. During the SLN development process, response surface methodology was exploited to optimize the composition of solid lipid and surfactant. Optimum combination of Biogapress Vegetal 297 ATO and Tween 80 was found essential to produce SLNs of 134 nm. The oral bioavailability study showed that acyclovir-loaded SLNs possessed superior oral bioavailability when compared with the commercial acyclovir suspension. The plasma concentration of acyclovir-loaded SLNs was four-fold higher than the commercial suspension. Thus, this investigation presented promising results that the method developed for encapsulation of acyclovir offers potential as an alternative pathway to enhance the drug’s bioavailability. In conclusion, this study exhibited the feasibility of SLNs as an oral delivery vehicle for acyclovir and therefore represents a new promising therapeutic concept of acyclovir treatment via a nanoparticulate drug delivery system.

## 1. Introduction

*Herpes simplex* virus (HSV) infection constitutes a significant health and financial burden globally [1,2]. Most recently, the World Health Organization (WHO) reported that the prevalence of HSV Type 2 infection was about 491 million aged up to 49 years globally in 2016, while oral HSV Type 1 has infected about 3583.5 million people, which is more common than the genital HSV Type 1 infection, estimated around 122–192 million cases [2]. Both orofacial and genital HSV have caused serious wide-spread infection in developed and developing countries, and more than one third of the population has clinically recurrent infections [2,3,4].

The current treatment option for HSV infections is prescription of an antiviral drug, acyclovir [5,6]. Acyclovir also known as acycloguanosine is a guanosine analog with an attachment of aliphatic group on the side chain (9-[2-hydroxyethoxymethyl] guanine). This drug is efficient against most species of the herpes virus family and actively absorbed in the upper gastrointestinal tract (GIT), particularly the duodenum as well as the jejunum. However, the absorption and oral bioavailability of acyclovir are very low, in the range 15–30%. Approximately, 70–80% of the oral dose is not absorbed and excreted via feces. Moreover, the terminal half-life of acyclovir is short, about 2.5–3 h following drug administration [7,8].

Due to these unfavorable characteristics, patients are required to take frequent and high doses of acyclovir (200 mg, five times daily, up to ten consecutive days) to achieve therapeutic efficacy. However, high doses of acyclovir have been shown to result in side effects including acute kidney injury [9,10,11]. Formation of acyclovir crystals in the renal duct (crystalluria) may occur within 24–48 h following acyclovir therapy and can cause blockage of nephrons, edema, bleeding of the renal interstitium and reduced glomerular filtration rate with concomitant kidney injury [12,13]. Therefore, a suitable drug carrier system that can easily be loaded with acyclovir and overcome its pharmacokinetic limitations should be developed.

Solid lipid nanoparticles (SLNs) have shown immense potential as drug carriers and have popularly been adopted as excellent candidates to fulfill the goal of improving absorption and oral bioavailability of certain drugs. SLNs represent one of the latest colloidal carriers developed for pharmaceutical formulation and drug delivery systems as an alternative to the existing conventional drug carriers such as emulsions and liposomes. This new generation of nanoparticles has attracted major attention with potential application not only in the pharmaceutical field but also in other commercial industries such as cosmetics, worldwide. This system provides excellent biocompatibility with high tolerability, better physicochemical stability (long-term stability) and controlled or sustained drug release property [14,15,16,17,18].

Recently, several studies have attempted to develop suitable SLNs formulations for delivery of drugs via the oral route. SLNs fabricated from various types of lipids function as good absorption enhancers in the gastrointestinal tract and assist in elevating the oral bioavailability of drugs. Besides, solid lipids that are used to formulate SLNs are generally recognized as safe, and often exhibiting very minimal or no toxicity [15,19,20,21,22]. It has been suggested that incorporation of drugs into SLNs might alter their intestinal permeability following oral administration. Drugs that are encapsulated into SLNs could gain access to the systemic circulation via the lymphatic system through M cells of Peyer’s patches found in the ileum [23,24,25].

Thus far, information regarding the application of nanoparticles for enhancement of oral bioavailability of acyclovir is very limited. Therefore, the main objective of this study was to investigate the influence of SLNs on the oral bioavailability of acyclovir. It is hypothesized that acyclovir-loaded SLNs would enhance the bioavailability when administered orally in an in vivo model. Encapsulation of acyclovir into SLNs would improve the efficacy of this drug as first line treatment of herpes infections; thereby, lower doses of drug would be sufficient to provide similar therapeutic effects as the commercial oral suspension. Patients may be greatly benefited from the low dosage regimen as it can reduce the risk for kidney injury considering acyclovir is usually used over long period of time for prophylaxis of HSV infections. In this study, SLNs formulation was developed using a plant-based solid lipid, glyceryl palmitostearate (Biogapress Vegetal 297 ATO). The solid lipid nanoparticles were then further characterized and subjected for thermal analysis, in vitro drug release and in vivo pharmacokinetic parameters of the acyclovir-loaded nanoparticles system were also evaluated. To date, to our knowledge, this is the first study exploring the potential of Biogapress Vegetal 297 ATO-based SLNs in improving the pharmacokinetic profile of orally delivered acyclovir.

## 2. Materials and Methods

### 2.1. Materials

Acycloguanosine (Acyclovir, 99% pure compound) and polysorbate 80 (Tween 80) were purchased from Sigma-Aldrich (St. Louis, MO, USA). Glyceryl dipalmitostearate, Biogapress Vegetal 297 ATO were supplied as a gift by Gattefosse (Lyon, France). Deionized water was collected using Milli-Q filtration system (Merckmillipore, MA, USA). All chemicals used in this study were of the highest purity grade available and prepared as instructed.

### 2.2. Central Composite Design

Central Composite Design (CCD) was employed to explore the effect(s) of independent variables and composition of Biogapress Vegetal 297 ATO and Tween 80 on the physical characteristics of SLNs, namely particle size, zeta potential and polydispersity index. The upper and lower limits of the independent variables are illustrated in Table 1. Thirteen sets of experiments were generated by CCD through Design Expert software (version 6.0, Stat-Ease Inc., MN, USA) that comprised 5 replicates of center points, 4 axial and 4 factorial points and was carried out in randomized order. All input data were analyzed by response surface regression procedure and polynomial model was chosen based on the significant terms (*p* < 0.05). The least significant lack of fit, coefficient of variance and multiple correlation coefficient were provided by Design Expert software.

### 2.3. Statistical Analysis

Optimum composition of lipids and surfactants (independent variables) were chosen to obtain minimum particle size, maximum zeta potential and minimum polydispersity index. The response surface behavior was investigated for the response function (*y*) using the polynomial equation and the generalized response surface model as shown below.
(1)*y* = *β*_0_*x*_0_ + *β*_1_*x*_1_ + *β*_2_*x*_2_ + *β_k_x_k_*
where *y* is the predicted response. *β*_0_ is a constant. *β*_1_*, β*_2_ and *β_k_* are the linear, quadratic and interaction coefficients, respectively.

Analysis of variance (ANOVA) was used to determine significant differences between the independent variables. Significant independent variables (*p* < 0.05) were included in the reduced model. Three-dimensional response surface plots were also constructed to visualize the interaction between variables. For models with a good fit, *R*^2^ value should be minimum of 0.8.

### 2.4. Verification of the Models

Student’s *t*-test was carried out between the theoretical prediction and experimental values obtained to validate the established models. *p* < 0.05 was considered as significant.

### 2.5. Preparation of Solid Lipid Nanoparticles

SLNs were prepared using hot high-shear homogenization and ultrasonication method. Solid lipid and aqueous phase consisting of deionized water and Tween 80 were pre-heated to 65 °C and combined to form a coarse emulsion while magnetically stirring at 600 rpm. For acyclovir-loaded SLNs, 10 mg of acycloguanosine were dispersed into molten lipid and the resulting formulation end weight was 20 g. A high-shear homogenizer (Ultra-Turrax T25, IKA, Staufen, Germany) set at 20,000 rpm for 5 min was used to emulsify the mixture, followed by a 10-min ultrasonication procedure at 80% intensity. The formulation was left at 25 °C for at least an hour to form SLNs and stored in amber-colored bottles for further analyses.

### 2.6. Size, Zeta Potential and Polydispersity Index (PdI) Analysis

A particle size analyzer (Malvern Nano ZS90, Malvern, UK) was used to measure the average size, zeta potential and PdI of SLNs. All samples were diluted (1:100) with deionized water prior to analysis to ensure a suitable scattering intensity and placed in cuvettes for measurement. Data were generated at 25 °C at a fixed 90° light incidence angle. All measurements were acquired by calculating the average of 10 runs for all independent preparation of blank and acyclovir-loaded SLNs samples (*n* = 3).

### 2.7. Drug Entrapment Efficiency (EE)

The concentration of free drug from SLNs was determined using ultrafiltration-centrifugation technique to calculate the percentage entrapment efficiency (*%EE*). Three independent preparation of acyclovir-loaded SLNs samples were filtered through centrifugal filter devices and centrifuged at 5000 rpm for 10 min with a fixed 23° angle rotor. Concentration of acyclovir presented in the supernatant (unentrapped) was quantified using a UV spectrophotometer at 254 nm. The formula adopted for EE calculations was as follows:$$\%EE=\;\frac{Total\;amount\;of\;drug-Unentrapped\;drug}{Total\;amount\;of\;drug}\;\times \;100$$
where *total amount of drug* is the amount of drug added into the lipid phase and *unentrapped drug* is the estimated amount of drug presented in the aqueous phase of the formulation measured after centrifugation and filtration of the samples.

### 2.8. Transmission Electron Microscopy (TEM)

A transmission electron microscope (H-7100 Hitachi Ltd., Tokyo, Japan) was used to visualize SLNs. Firstly, samples were diluted 1:100 in deionized water and placed on a 3-mm carbon-coated copper grid. Uranyl acetate solution 2% (*w/v*) was used to stain the samples and grids were left to dry at 25 °C prior to microscopic observation.

### 2.9. Differential Scanning Calorimetry (DSC)

A differential scanning calorimeter, DSC822e instrument (Mettler Toledo, Greifensee, Switzerland), was used for thermal analysis. Each sample was accurately weighed on a plate and sealed non-hermetically. An empty standard aluminum pan was used as a reference, and purge gas, helium, was supplied at a rate of 50 mL/min. The melting point, onset temperature of endothermic drop and melting enthalpy of SLNs were calculated by the software provided. The recrystallization index (RI) of SLNs formulation was also calculated using the equation below:$$RI\left(\%\right)=\;\frac{\Delta H\;SLNs\;\left(J/g\right)}{\Delta H\;bulk\;material\;\left(J/g\right)\;\times \;Concentration\;lipid\;phase\;\left(\%\right)}\;\times \;100$$
where ∆*H SLNs* is the melting enthalpy of the SLN dispersion and ∆*H bulk material* is the melting enthalpy of bulk lipid.

### 2.10. In Vitro Release Study

The in vitro drug release study was carried out in enzyme-free simulated intestinal fluid (SIF) at pH 6.8, prepared according to United States Pharmacopeia (USP). This setting was suitable to assess the dissolution and release profile of acyclovir in simulating upper intestinal environment for absorption process in small intestine. Acyclovir-loaded SLNs or commercial suspension (equivalent to 5 mg of acyclovir) were loaded into a dialysis bag (cellulose membrane, molecular weight cut-off of 12,000 Da, Sigma Aldrich, St. Louis, MO, USA), soaked in 50 mL of release media and magnetically stirred at 100 rpm while being maintained at 37 ± 0.5 °C throughout the study. Release media (5 mL) was withdrawn from the beaker at selected time intervals (0.5, 1.0, 1.5, 2.5, 3.0, 4.0, 5.0, 6.0, 8.0, 10.0, 12.0 and 24 h) and an equal amount of fresh release media was added to maintain sink condition. Concentration of acyclovir in the withdrawn media was determined using a UV spectrophotometer set at 254 nm (UV-1800 UV-VIS Shimadzu, Kyoto, Japan). The release profile of acyclovir-loaded SLNs was compared with acyclovir commercial suspension. All experiments were performed in replicate and results are reported as average ± standard deviation (S.D.) of three independent preparations of each formulation (*n* = 3).

### 2.11. In Vivo Pharmacokinetic Evaluation

#### 2.11.1. Animal Study

Twelve male Sprague-Dawley rats weighing between 200 and 250 g were obtained from a supplier, Takrif Bistari Enterprise (Seri Kembangan, Selangor, Malaysia). Rats were acclimatized for one week prior to the experiment and maintained on a 12-h light/dark cycle with controlled temperature (24 ± 2 °C) and relative humidity (60 ± 5%). All animals were housed individually in standard laboratory polypropylene/polycarbonate rat cages (435 mm × 290 mm × 150 mm) with access to food and water ad libitum. Animals were randomly divided into two groups (*n* = 6) and administered either acyclovir-loaded SLNs (equivalent to 20 mg/kg of acyclovir) or 20 mg/kg of commercial acyclovir suspension per orally using an oral feeding (gavage) needle.

#### 2.11.2. Blood Sample Collection and Plasma Preparation

About 300–400 µL of blood were withdrawn from the tail vein and deposited into heparinized microcentrifuge tubes at designated time intervals post oral administration: 0, 0.5, 1.0, 2.5, 4.0, 6.0, 10.0 and 24.0 h. All samples were centrifuged for 15 min at 2000 *g* and supernatant (plasma) was collected. All plasma samples were kept at -20 °C freezer prior to analysis.

#### 2.11.3. Ultra Performance Liquid Chromatography (UPLC)

Acquity ultra performance liquid chromatographic system (Waters, MA USA) with UV detection set at 254 nm was used to quantify the concentration of acyclovir in plasma as described by previous studies [26,27]. UPLC was equipped with a photodiode array (PDA) detector and a quaternary solvent delivery system. Data obtained were processed using the chromatographic software, Empower 3 (Waters, MA, USA). Analysis was performed using an Acquity BEH C18 (100 × 2.1 mm, 1.7 µm) column (Waters, MA, USA) at room temperature (25 °C). The mobile phase for acyclovir separation was a mixture of 0.02-M potassium dihydrogen phosphate and acetonitrile at ratio 97:3 with a final pH of 2.5. The mobile phase flow rate was set at 0.20 mL/min. Plasma samples (10 μL) were injected into the UPLC system with four minutes of total run time for each sample.

#### 2.11.4. Plasma Protein Precipitation Procedure for Determination of Acyclovir Concentration

Prior to UPLC analysis, frozen plasma samples were thawed for 10 min at room temperature. For protein precipitation, 5% perchloric acid was added to each sample at a ratio of 1:1 and vortexed for 30 s followed by 10,000 rpm centrifugation at 4 °C for 10 min. The collected supernatant was filtered through a 0.45-µm nylon syringe filter and subsequently injected into the UPLC system.

### 2.12. Pharmacokinetic Parameters

The maximum drug concentration observed in plasma (*C_max_*) and time to reach maximum concentration (*T_max_*) were observed directly from the plasma acyclovir concentration versus time point plot. The elimination constant (*K_e_*) and half-life (*T*_1/2_) were determined from the elimination phase of the concentration versus time profile. The area under the curve calculated up to 24 h (*AUC*_0–24_) and area under the curve calculated to infinity (*AUC*_0–∞_), which represent the total amount of acyclovir that reaches the systemic blood circulation for all plots, were also evaluated. A linear-log trapezoidal technique was employed to calculate the *AUC*. The relative bioavailability of acyclovir in the plasma samples administered orally were calculated using the formula below:$$Relative\;Bioavailability=\;\frac{{\left[AUC\right]}_{A}*\;dose_{B}}{{\left[AUC\right]}_{B}*\;dose_{A}}\;\times \;100$$
where [*AUC*]*_A_* is area under the curve of the test formulation and [*AUC*]*_B_* is area under the curve of the reference formulation.

### 2.13. Statistical Analysis

Statistical analyses were performed using Prism GraphPad software (GraphPad, CA, USA). Pharmacokinetic parameters obtained from the experiments were expressed as mean ± standard deviation (S.D.). Student *t*-test was used to evaluate the statistical significance. A *p*-value of less than 0.05 (*p* < 0.05) was considered as statistically significant.

## 3. Results and Discussion

### 3.1. Fitting the Response Surface Methodology

The experimental data acquired for all response variables based on CCD matrix were used to determine the model of best fit for the independent variables of SLNs. All equations showed small p-value (*p* < 0.05) and suitable coefficient of determination (*R*^2^ > 0.9), which indicate that the quadratic polynomial model is highly significant and sufficient in representing the actual interaction between the independent variables and the responses (Table 2).

In the equation, non-significant (*p* > 0.05) linear terms were also included in the final reduced model if the quadratic or interaction terms containing these variables were significant (*p* < 0.05). The positive value in the equation reflects the synergistic effect that favors the optimization process. This showed that the value of responses would increase with an increment in the level of the independent variables. On the other hand, the negative value is an indicator of antagonistic effect between the factors and responses [28]. Effect of the amount of solid lipid on particle size presented positive signs, which indicated that increasing amount of solid lipid led to formation of larger particle size.

In contrast, the effect of amount of surfactant showed negative sign that indicates an increasing amount of surfactant in SLNs formulation would lead to formation of smaller sized particles. From the data, both solid lipid and surfactant showed a negative impact on the polydispersity index. For zeta potential, solid lipid showed negative impact, but surfactant had a positive impact, indicating that increasing surfactant amount would lead to lesser negative charge of particle surface.

Evaluation of the significance of quadratic polynomial models was carried out using Analysis of Variance (ANOVA) (Table 3). Significant effect on the response variables of any terms in the models was represented by large F value and small p-value (*p* < 0.05). The results demonstrate that interaction between solid lipid and surfactant had significant effect on all responses observed in the study.

### 3.2. Response Surface Analysis

Response surface plots were employed to illustrate the effect of interaction between independent variables, solid lipid and surfactant, on particle size (Figure 1). From the plot, it was observed that increment of lipid concentration caused an increase in particle size (Figure 1a). A few factors should be taken into consideration when describing this observation. Firstly, high lipid content would result in increment of viscosity and flow resistance of the hot oil droplet. Thus, disruption process would be difficult and might restrict the breakup rate of the oil droplet, hence forming larger sized particles [28,29]. Secondly, if the surfactant molecules present in the aqueous phase were insufficient to cover the surface of the nanoparticles and the collision rate of SLNs formed was faster than the rate at which surfactant molecules were adsorbed onto the particles, particle aggregation and agglomeration could occur due to hydrophobic interaction between the particles [30]. Further addition of Tween 80 in the aqueous phase up to the optimum concentration (3.0% *w/w*) would reduce the size of particles formed. In other words, optimum amount of surfactant is crucial to produce smaller particle size due to availability of surfactant to adsorb onto the newly formed surfaces.

Increment of surfactant concentration at lower composition of Biogapress Vegetal 297 ATO (i.e., 300 mg) was shown to reduce the zeta potential value, which became less negative (Figure 1b). These data are in agreement with a previous study where a similar observation was reported for SLNs stabilized using Tween 80 [31]. The densely packed and thicker layer of non-ionic surfactant such as Tween 80 that adsorbed onto the surface of SLNs produced steric stabilization (hindrance) instead of electrostatic repulsion and shifted the slipping plane (where zeta potential is measured) away from the surface of the particles, causing lower value of zeta potential measurement [32]. Therefore, it was suggested that a decrease in zeta potential measurement might not be an indicator of reduction in SLNs stability.

A higher surfactant amount also led to an increase in PdI value that could be explained by an increase in number of non-uniform micelles/particle shape and size formation [33]. High viscosity of aqueous phase due to presence of a higher concentration of surfactant has affected the emulsification efficiency during SLNs preparation. As a result, particles with varying sizes were formed that contributed to higher PdI. Therefore, the data suggest that smaller particle size and lower PdI could be produced using optimum concentration of solid lipid and surfactant formulations.

### 3.3. Verification of the Reduced Model

The experimental and predicted values from the response surface model for all responses deduced from the suggested final composition of solid lipid and surfactant were compared to validate the accuracy of the response surface equations generated by RSM (Table 4). Data show no significant difference (*p* < 0.05) between the experimental and predicted values for all responses. Thus, it was concluded that the response surface models were proven and verified as there were good concurrences and agreement between the experimental and predicted values.

### 3.4. Physical Characteristics, Morphology and Entrapment Efficiency of Acyclovir-Loaded SLNs

The average particle size of three independent preparations of SLNs loaded with acyclovir was 123.70 ± 6.47 nm, as measured by a particle size analyzer. The results of the study indicate that incorporation of acyclovir did not exert significant influence on the physical characteristics of the SLNs system. A similar observation was reported previously, where incorporation of different amounts of risperidone in SLNs did not significantly influence the particle size of freshly prepared formulations using ultrasonication method measured during the production day [34]. Earlier studies reported that maintaining small particle size of SLN dispersions for its oral delivery is important because particle sizes of less than 400 nm (submicron) are suitable for gastrointestinal absorption [35,36]. The size of SLNs was found to be consistent between dynamic light scattering method (zetasizer) and observation under TEM. The images of TEM (Figure 2) captured showed SLNs were spherical in shape with smooth surface and uniformity similar to other studies [29,33,37].

PdI is a measurement of the broadness of particle size distribution and the quality of SLNs system. PdI values that are closer to 0 (PdI values of ≤ 0.1) show excellent particle distribution where the particles are said to be in monodisperse or uniform distribution [38]. From the data obtained in this study, the average PdI values measured for three independent preparation of acyclovir-loaded SLNs samples was 0.22 ± 0.02, suggesting a low value of PdI. This value indicated that SLNs were uniformly distributed, where the size and shape of the particles were consistent (homogeneous particle size distribution) [34,39]. Usually researchers consider PdI values of less than 0.5 as acceptable while 0.3 and below are regarded as optimum PdI values.

Zeta potential is another important parameter in evaluating the colloidal dispersion quality and its probable physical stability. The zeta potential value can be either positive or negative contributed by the electrical potential produced by the charge presented on the surface of each particle. It also denotes the magnitude of electrostatic repulsion between particles of similar charges in the aqueous dispersion [40,41]. The average zeta potential value for three independent preparation of acyclovir-loaded SLNs samples tested was −27.20 ± 2.53 mV, indicating that the incorporation of acyclovir into SLNs systems had little effect and did not significantly change the zeta potential value of both SLN dispersions measured. This is in agreement with previous reports where incorporation of drugs into SLNs did not alter the zeta potential value of the system [39,42].

The average percentage of drug entrapment efficiency for three independent preparation of acyclovir-loaded SLNs was relatively high, 86.53 ± 0.49%. Drug entrapment efficiency of SLNs could be associated with the type of solid lipid used in the formulation. It was reported previously that solid lipids made up of pure triglycerides such as trilaurin had lower entrapment efficiency and encountered drug expulsion. This was due to rigid, solid and perfect structural arrangement of the lipid matrix that produced limited space to accommodate drug molecules [40]. In contrast, a more complex lipid mixture of monoglycerides, diglycerides and triglycerides would theoretically form irregular or less ordered arrangement of crystalline structure (lipid matrices). Thus, ample space is available to incorporate drug molecules and eventually permit a higher percentage of drug entrapment [40]. In this study, the solid lipid used to fabricate SLNs, Biogapress Vegetal 297 ATO, is comprised of 33% monoglycerides, 37% diglycerides and 30% triglycerides based on technical data sheets from the manufacturer, Gattefosse. Considering this fact, it was presumed that Biogapress Vegetal 297 ATO SLNs had irregularities in the crystalline lattice, hence providing sufficient room for encapsulation of acyclovir into the system. To confirm this statement, differential scanning calorimetry analysis was performed.

### 3.5. Differential Scanning Calorimetry Analysis

The DSC analysis of acyclovir pure compound demonstrated a single melting peak at 249.3 °C (Figure 3a), almost identical to the melting temperature provided in the clinical data sheet by the manufacturer, Sigma Aldrich (255 °C). This indicated that the drug remained stable during preparation of SLNs using hot high-sheer homogenization technique at temperature of 65 °C. Therefore, this preparation method is appropriate and should be proposed for production of effective SLNs to encapsulate drugs with higher melting temperature than the solid lipids.

Different lipid modifications possess distinct melting enthalpy and melting points due to their crystalline nature. DSC analysis was employed to determine alteration in the thermal state of the solid lipid (Figure 3). Bulk powder of Biogapress Vegetal 297 ATO produced a single melting peak with maximum temperature of 64.93 °C. When solid lipid was fabricated as nanoparticles, blank SLNs and acyclovir-loaded SLNs exhibited lower melting points of 56.13 and 56.37 °C, respectively. The melting enthalpy of SLNs also decreased to 3.93 (blank) and 4.34 J/g (acylovir-loaded) when compared with their bulk counterpart (207.95 J/g). The degree of crystallinity (% RI) was calculated and bulk lipid was fixed as 100% (fully crystalline), a reference to establish the degree of SLNs crystallinity. The RI of blank and acyclovir-loaded SLNs reduced to 37% and 41%, respectively.

Lower melting temperature, onset temperature and melting enthalpy of blank and acyclovir-loaded SLNs, as seen in the thermograms, were due to less-ordered lipid structure of SLNs, which required lesser energy to break the lattice forces when compared with the highly crystalline bulk lipid [39,43]. Consequently, onset temperature and melting point of the SLNs were shifted to a lower temperature. In addition, the calculated degree of crystallinity or RI of SLNs supported these findings. Hence, the SLNs were proven to possess less-ordered crystal matrix, a fundamental characteristic of SLNs as a drug delivery vehicle. Solid lipid with lower level organization of crystal lattice exhibited a higher percentage of entrapment efficiency and/or controlled release of drug [44,45]. It is acceptable that the aforementioned theory is in agreement with the present data where high percentage of acyclovir entrapment efficiency was observed.

Most importantly, the maximum peak temperature that resembles the melting point of SLNs observed in this study was still higher than 50 °C. Melting temperature of the fabricated SLNs should remain higher than room temperature (25 °C) and body temperature (37 °C) to maintain their solidity during oral administration. Additionally, if the melting temperature of SLNs is below room temperature, supercooled melts (mixture of emulsions) could be generated rather than formation of SLN dispersions [34,46]. Therefore, from the thermal analysis performed in this study, we are convinced that the SLNs developed were in their solid state.

### 3.6. In Vitro Release Study

The commercial acyclovir suspension (*n* = 3) showed 100% cumulative drug release within 2 h, while three independent preparation samples of acyclovir-loaded SLNs had biphasic release profile. During the initial state, about 46% of the drug was released from acyclovir-loaded SLNs within 1 h. Then, a sustained release pattern was observed until it reached 100% cumulative release at the 24 h (Figure 4). The burst release during the first hour of the in vitro release study was due to the presence of acyclovir molecules that were entrapped on the SLNs surface. Formation of SLNs with drug rich shell type was commonly reported in association with the application of hot homogenization method during fabrication process. High temperature was applied during hot homogenization process, causing partitioning of drug molecules into aqueous phase, to leave only small amount of drug in the lipid melt. When cooling phase took place and initial lipid solidification ensued, re-partitioning of drug molecules into lipid phase occurred. However, during this cooling phase, drug molecules could no longer access the solidified pure lipid core and become entrapped in the outer layer of the still-melted solid lipid. As a consequence, drug rich shell model was produced with shorter pathway for drug to be released from the SLN matrices [33].

### 3.7. In Vivo Oral Bioavailability and Pharmacokinetic Evaluation

The oral bioavailability and pharmacokinetic parameters of acyclovir-loaded SLNs formulation subsequent to its administration in vivo were calculated while commercial acyclovir suspension was used as a reference formulation. The time to reach maximum acyclovir plasma concentration for acyclovir-loaded SLNs treatment group (*n* = 6) was 1.25 h while commercial acyclovir suspension reported at the first hour post-oral administration (Figure 5). This observation could be due to the burst release effect of acyclovir-loaded SLNs within the first 2 h after their oral administration, in agreement with the in vitro drug release data described above.

The findings of this study revealed that the half-life of commercial acyclovir suspension did not differ from previous studies, ranging 2–3 h [27,47]. The half-life of acyclovir when incorporated into SLN dispersions was extended up to 5 h, similar to the findings reported in an in vivo study of acyclovir-loaded PLGA nanoparticles [48]. Experimental results also show acyclovir-loaded SLNs had extended release profile of up to 24 h with high plasma concentration of acyclovir corresponding to the in vitro drug release data, as discussed above. However, concentration of acyclovir from the commercial suspension was undetectable in plasma beyond 10 h post oral acyclovir administration.

Data from the in vivo pharmacokinetic study show SLNs significantly (*p* < 0.05) enhanced acyclovir absorption when compared with commercial oral acyclovir suspension. A significant improvement (*p* < 0.05) in *AUC*_0–24_ and *AUC*_0–∞_ of acyclovir plasma concentration was observed for acyclovir-loaded SLNs that was approximately 4.23 times higher than the commercial acyclovir suspension (Table 5). The relative bioavailability of acyclovir-loaded SLNs following oral administration was also calculated to estimate fraction of free drug available in the systemic circulation from a test formulation relative to the reference formulation (a formulation that is commercially available on the market) of a drug. It was found that acyclovir-loaded SLNs had 423.61% relative bioavailability, supporting the hypothesis that SLNs as acyclovir carriers enhanced its oral bioavailability.

Acyclovir is classified as a compound with high solubility but low or limited intestinal permeability, a class III category according to the Biopharmaceutical Classification System (BSC). A study showed that a limited fraction of acyclovir was absorbed into the systemic circulation due to its poor intestinal permeability [49]. Similarly, low plasma concentration of commercial acyclovir was observed in this study, with calculated *AUC*_0–∞_ of 1341.67 ± 133.40 h·ng·mL^−1^. In comparison to the above data, results from rats administered with acyclovir-loaded SLNs showed significant and superior increment in the *AUC*_0–24_ and *AUC*_0–∞_ (Table 5). Several factors were suggested in previous studies to elaborate the possible mechanism(s) underlying enhancement of the oral bioavailability of a drug when loaded into nanocarriers. For instance, studies reported that SLNs could enter the lymphatic circulation and are further transferred to systemic circulation via two possible mechanisms: (i) microfold cells (M cells) uptake nanoparticles in the Peyer’s patches; and (ii) paracellular and/or transcellular uptake SLNs [22,25,50].

Apart from that, previous studies demonstrated that acyclovir is a P-glycoprotein substrate [51,52]. P-glycoprotein is located at the membrane of enterocytes that facilitates efflux of drug compounds from inside epithelial cells back to the intestinal lumen to limit the transport of drug molecules into the systemic circulation or prevent drug absorption, causing oral bioavailability and concentration of the drug to be at the lowest level [53]. It was proposed that inhibition of P-glycoprotein could be a potential approach to alter the pharmacokinetics and bioavailability of acyclovir. Researchers have predicted that administration of drugs encapsulated in nanoparticles via oral route could induce inhibition of the P-glycoprotein efflux transporter and enhance drug permeability and its subsequent oral absorption [36,54]. This could be associated with the presence of surfactants such as Tween 80 or Cremaphore EL in the nanoparticle formulation. These surfactants were found to be inhibitors of P-glycoprotein apart from being employed in pharmaceutical formulations as agents that increase the intestinal membrane fluidity [55,56]. In this study, acyclovir-loaded SLNs showed superior oral bioavailability in comparison to the conventional acyclovir suspension. Hence, it was surmised that this observation could be related to the inhibition or reduction in efflux activity of P-glycoprotein by Tween 80, which was used as a surfactant in the fabrication of the SLNs formulations. Furthermore, it was revealed in a previous study that in vitro intestinal transport and absorption of acyclovir increased by one-fold when the function of P-glycoprotein was inhibited, in comparison to the non-inhibition group treated with acyclovir alone [51]. Nevertheless, additional studies are warranted in the future to elucidate the mechanism(s) involved in acyclovir-loaded SLNs intestinal uptake and concomitant enhancement of acyclovir oral bioavailability. Apart from that, prior to the in vivo drug testing, it is very important to scrutinize the in vitro degradation of acyclovir-loaded SLNs and their in vitro cytotoxicity.

## 4. Conclusions

Based on the analysis and data obtained from the present study, the preparation, optimization, characterization and potential application of SLNs as a nanoparticulate acyclovir carrier were well explored. It is recommended that SLNs formulated from Biogapress Vegetal 297 ATO could be suitable oral drug delivery vehicles which can be adopted to overcome current pharmacological limitations of acyclovir therapy. Moreover, Biogapress Vegetal 297 ATO is a plant-derived pharmaceutically safe material. The experimental works documented present significant contributions to the area of oral drug delivery systems in general. Beneficial and important information regarding the development of SLNs as a new strategy and alternative acyclovir delivery vehicle to improve its oral bioavailability was attained and this can be a breakthrough in the efforts towards combating HSV infections. This discovery may serve as a platform for development of more efficacious oral drug delivery systems in general.

## Figures and Tables

**Figure 1 nanomaterials-10-01785-f001:**
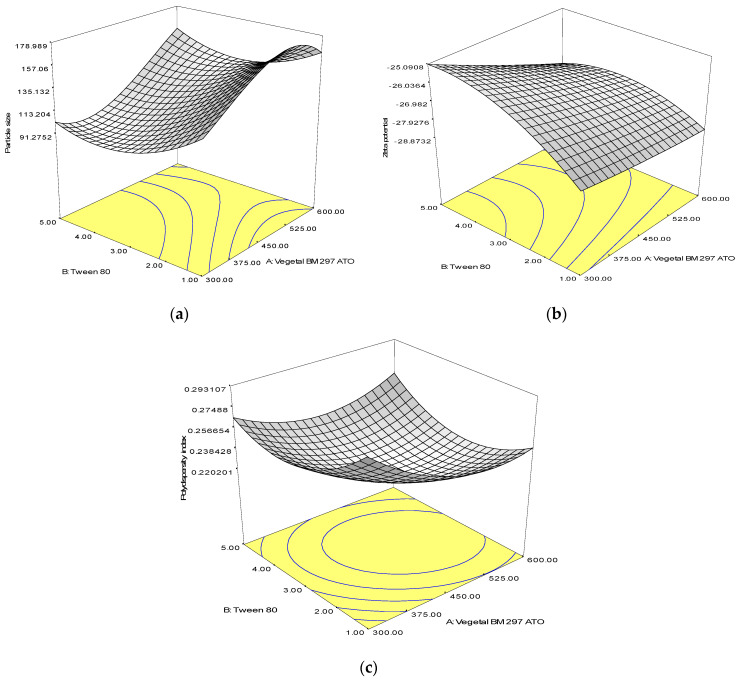
Response surface plots showing the effect of interaction between Biogapress Vegetal 297 ATO and Tween 80 compositions on: (**a**) particle size; (**b**) zeta potential; and (**c**) polydispersity index.

**Figure 2 nanomaterials-10-01785-f002:**
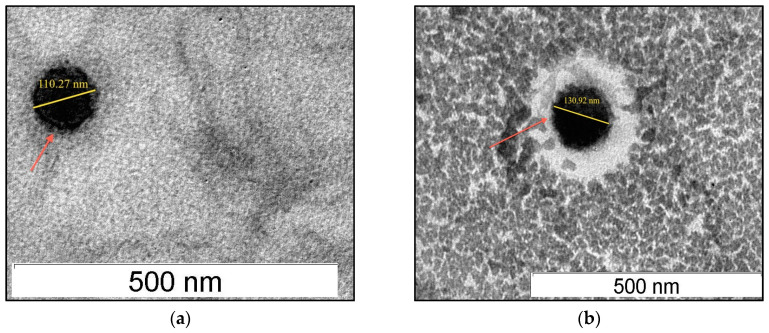
Transmission electron micrograph at 60,000× magnification of: drug-free SLNs (**a**); and acyclovir-loaded SLNs (**b**).

**Figure 3 nanomaterials-10-01785-f003:**
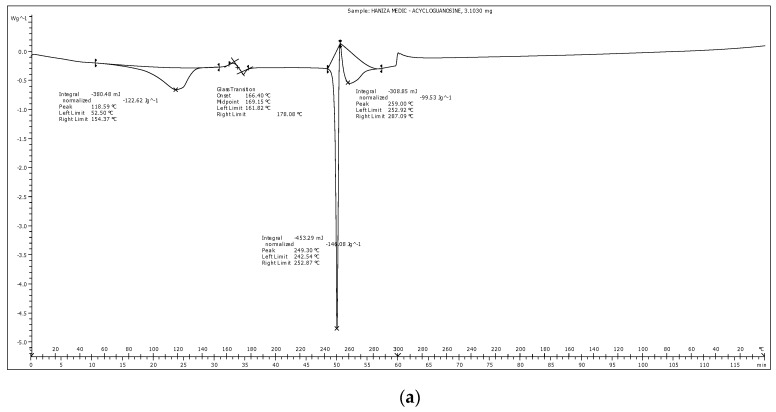
DSC thermograms of: (**a**) acyclovir pure compound and (**b**) Biogapress Vegetal 297 ATO bulk powder (VEGETAL BULK POW), SLNs loaded with 10 mg of acyclovir (VO20) and drug-free SLNs (VEGETAL BLANK).

**Figure 4 nanomaterials-10-01785-f004:**
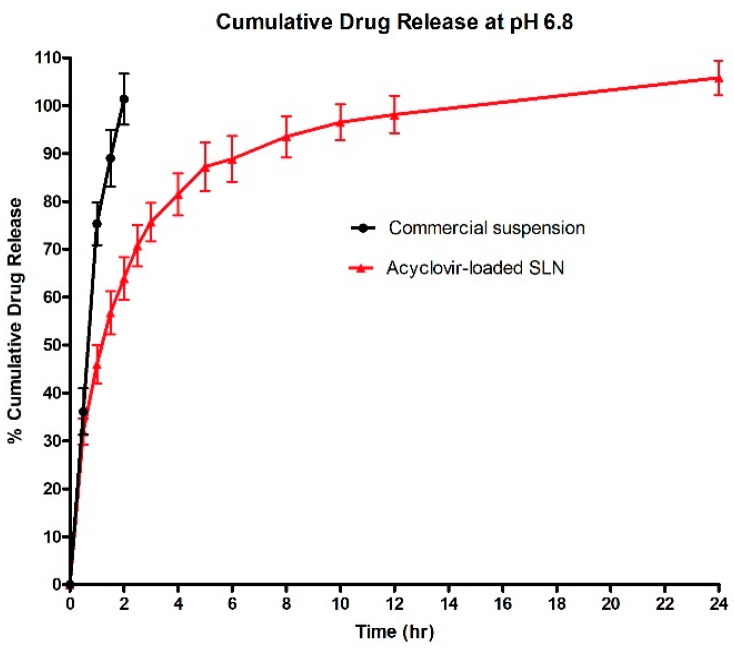
Cumulative percentage of acyclovir release profiles from acyclovir-loaded SLN dispersions and commercial oral suspension at pH 6.8. The data represent mean ± S.D. (*n* = 3).

**Figure 5 nanomaterials-10-01785-f005:**
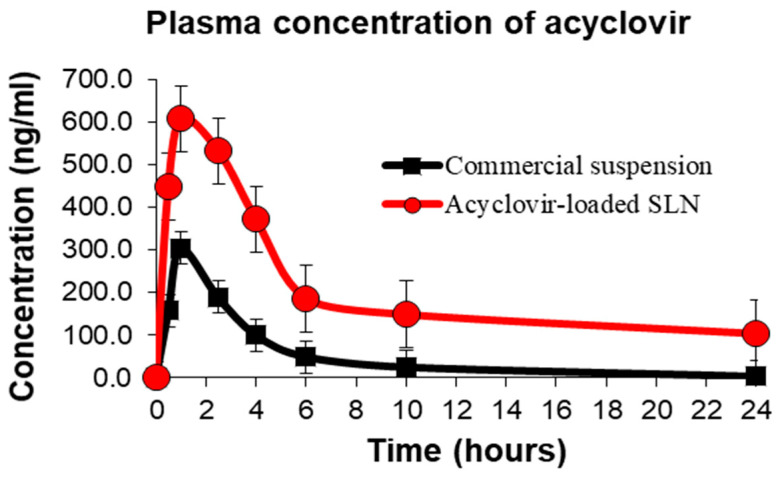
Plasma concentration versus time profile after oral administration of acyclovir-loaded SLNs or commercial suspension in rats. The data represent mean ± S.D. (*n* = 6).

**Table 1 nanomaterials-10-01785-t001:** Independent variables with high and low levels.

Independent Variables	Coded Levels				
	Axial(−*α*)	Low	Centre	High	Axial(+*α*)
Biogapress Vegetal 297 ATO (mg)	237.87	300.00	450.00	600.00	662.13
Tween 80 (*w/w*)	0.17	1.00	3.00	5.00	5.83

**Table 2 nanomaterials-10-01785-t002:** Equations, regression coefficients and probability values for reduced model of SLNs.

Size, *R*_1_	Equation: 118.82 + 33.92*A* − 32.74*B* − 1.82*A*^2^ + 20.85*B*^2^ + 5.13*AB* *R*^2^ value: 0.9992 *p*-value: <0.0001
Zeta Potential, *R*_2_	Equation: −26.78 − 0.93*A* + 0.96*B*+0.084*A*^2^ − 0.87*B*^2^ − 0.58*AB* *R*^2^ value: 0.9492 *p*-value: 0.0002
Polydispersity Index, *R*_3_	Equation: 0.22 − 0.01*A* − 0.004*B* + 0.02*A*^2^ + 0.025*B*^2^ + 0.01*AB* *R*^2^ value: 0.9460 *p*-value: 0.0003

A, Biogapress Vegetal 297 ATO; B, Tween 80 composition.

**Table 3 nanomaterials-10-01785-t003:** ANOVA of regression coefficient of the fitted quadratic equation for SLNs.

	Variables	Size		Zeta Potential		PdI	
		*F* Value	*p*-Value	*F* Value	*p*-Value	*F* Value	*p*-Value
**Main Effects**	*A*	575.69	<0.0001	26.16	0.0002	17.57	0.0041
	*B*	2681.96	<0.0001	42.55	0.0003	1.51	0.2592
**Quadratic Effects**	*A^2^*	14.45	0.0126	0.30	0.5990	43.01	0.0003
	*B^2^*	1892.21	<0.0001	32.44	0.0007	66.87	<0.0001
**Interaction Effect**	*AB*	65.72	0.0005	8.22	0.0241	6.03	0.0437

A, Biogapress Vegetal 297 ATO; B, Tween 80 composition.

**Table 4 nanomaterials-10-01785-t004:** The predicted and observed response value for the optimized SLNs. The observed data represent mean ± S.D. (*n* = 3).

Responses	Predicted	Observed
Particle Size (nm)	130.00	122.72 ± 2.15
Polydispersity Index	0.22	0.23 ± 0.01
Zeta Potential (mV)	−27.09	−24.37 ± 1.07

**Table 5 nanomaterials-10-01785-t005:** Pharmacokinetic parameters after oral administration of acyclovir-loaded SLNs or commercial suspension. The data represent mean ± S.D. (*n* = 6).

Parameters	Acyclovir Suspension	Acyclovir-Loaded SLNs
*C_max_* (ng/mL)	303.50 ± 26.70	607.00 ± 71.64 *
*T_max_* (h)	1.00 ± 0.00	1.25 ± 0.25
AUC_0-24_ (h·ng·mL^−1^)	1243.75 ± 125.90	4899.50 ± 321.30 *
*AUC*_0–∞_(h·ng·mL^−1^)	1341.67 ± 133.40	5683.43 ± 368.70 *
*K_e_* (h^−1^)	0.37 ± 0.05	0.14 ± 0.02
*T*_1/2_ (h)	2.06 ± 0.29	5.26 ± 0.55 *

* *p* < 0.05.

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
