# Peer review of "Acyclovir-Loaded Solid Lipid Nanoparticles: Optimization, Characterization and Evaluation of Its Pharmacokinetic Profile"

_nanomaterials, 2020, doi:10.3390/nano10091785_

Round 1
Reviewer 1 Report
Authors developed acyclovir-loaded solid nanoparticles and evaluate the formulation using in vitro release test and pharmacokinetic analysis.
However, some methods and results showed be revised.
- line 96-99, line 108-113, line 227-228. The variables in line 96-99 and Table 2 and equation symbol in the equation (1) should be unified otherwise it confused the readers.
- line 309-311. (Figure 2) was missing. The results showed single measurement data but authors described the particles size mean+/-SD without describing how they were calculated. Repeated measurement of single preparation or multiple independent preparation of acyclovir-loaded SLNs ?
- line 342. Similarly results of drug entrapment efficiency, RI, in vitro release test, and pharmacokinetic results should also be described as the number of experiments.
- Figure 4. DSC thermograms of Vegetel BLANK was missing. Acyclovir bulk powder should be added.
Author Response
- Line 106-107, the variables have been removed since they appeared in Table 2. The equation symbols in line 118-119 and 248-249 are unified.
- In method section, line 143, (n=3) was added. Line 146 and 185-186, ‘three independent preparation of acyclovir-loaded SLNs’ was added. In line 331-332, ‘three independent preparation of SLNs loaded with acyclovir was 123.70 ± 6.47 nm when measured using a particle size analyzer’ was included. In Table 4, number of independent samples was also included.
- Line 331, 346, 359, 422-423 number of experiments (three independent preparation of acyclovir-loaded SLNs) was included. Line 443, 449 and 460 number of animals (n=6) was added for pharmacokinetic results.
- Line 379, DSC of bulk acyclovir powder has been added in the manuscript (Figure 3 (a)) with description in line 396. The thermogram of vegetal blank is in blue line in Figure 3 (b).
Reviewer 2 Report
Please consider the indications from the attached document.

Author Response
- The novelty of the study was highlighted in line 83-93
“Encapsulation of acyclovir into SLNs would improve the efficacy of this drug as first line treatment of herpes infections, thereby, lower dose of drug is sufficient to provide similar therapeutic effects as the commercial oral suspension. Patients may be greatly benefited from the low dosage regimen as it can reduce the risk for kidney injury considering acyclovir is usually used over long period of time for prophylaxis of HSV infections. In this study, SLNs formulation was developed using a plant-based solid lipid, glyceryl palmitostearate (Biogapress Vegetal 297 ATO). The solid lipid nanoparticles were then further characterized and thermal analysis, in vitro drug release as well as in vivo pharmacokinetic parameters of the acyclovir-loaded nanoparticles system were also evaluated. To date, in our knowledge, this is the first study exploring the potential of Biogapress Vegetal 297 ATO-based SLNs in improving the pharmacokinetics profile of orally delivered acyclovir.”
2. Recent studies as references were added, as follow:
- Chih-Hung Lin, C.-H.C., Zih-Chan Lin, Jia-You Fang Recent advances in oral delivery of drugs and bioactive natural products using solid lipid nanoparticles as the carriers. Journal of Food and Drug Analysis, 2017. 25(2): p. 219-234.
- Rajpoot, K., Solid Lipid Nanoparticles: A Promising Nanomaterial in Drug Delivery. Curr Pharm Des, 2019. 25(37): p. 3943-3959.
- Jain, A., et al., Beta-carotene-Encapsulated Solid Lipid Nanoparticles (BC-SLNs) as Promising Vehicle for Cancer: an Investigative Assessment. AAPS PharmSciTech, 2019. 20(3): p. 100.
- Cirri, M., et al., Development and in vivo evaluation of an innovative "Hydrochlorothiazide-in Cyclodextrins-in Solid Lipid Nanoparticles" formulation with sustained release and enhanced oral bioavailability for potential hypertension treatment in pediatrics. Int J Pharm, 2017. 521(1-2): p. 73-83.
3. L171 changed to “prepared according to United States Pharmacopeia (USP)”
4. L174-176 (In vitro Release Study) – “This setting was suitable to assess the dissolution and release profile of acyclovir in simulating upper intestinal environment for absorption process in small intestine.” was added to justify pH 6.8
5. L 504-505, suggestion to study acyclovir SLNs in vitro degradation and more important SLNs in vitro cytotoxicity before in vivo testing was added in the discussion for future study.
6. References have been updated for examples:
James, C., et al., Herpes simplex virus: global infection prevalence and incidence estimates, 2016. Bull World Health Organ, 2020. 98(5): p. 315-329.
Alvarez, D.M., et al., Current Antivirals and Novel Botanical Molecules Interfering With Herpes Simplex Virus Infection. Front Microbiol, 2020. 11: p. 139.
Chavez-Iniguez, J.S., et al., Oral acyclovir induced hypokalemia and acute tubular necrosis a case report. BMC Nephrol, 2018. 19(1): p. 324.
Leowattana, W., Antiviral Drugs and Acute Kidney Injury (AKI). Infect Disord Drug Targets, 2019. 19(4): p. 375-382.
Chih-Hung Lin, C.-H.C., Zih-Chan Lin, Jia-You Fang Recent advances in oral delivery of drugs and bioactive natural products using solid lipid nanoparticles as the carriers. Journal of Food and Drug Analysis, 2017. 25(2): p. 219-234.
Rajpoot, K., Solid Lipid Nanoparticles: A Promising Nanomaterial in Drug Delivery. Curr Pharm Des, 2019. 25(37): p. 3943-3959.
Jain, A., et al., Beta-carotene-Encapsulated Solid Lipid Nanoparticles (BC-SLNs) as Promising Vehicle for Cancer: an Investigative Assessment. AAPS PharmSciTech, 2019. 20(3): p. 100.
Cirri, M., et al., Development and in vivo evaluation of an innovative "Hydrochlorothiazide-in Cyclodextrins-in Solid Lipid Nanoparticles" formulation with sustained release and enhanced oral bioavailability for potential hypertension treatment in pediatrics. Int J Pharm, 2017. 521(1-2): p. 73-83.
Ching-Yun Hsu, P.-W.W., Ahmed Alalaiwe, Zih-Chan Lin, and Jia-You Fang, Use of Lipid Nanocarriers to Improve Oral Delivery of Vitamins. Nutrients, 2019. 11(1):68: p. 30.
Ganesan, P., et al., Recent developments in solid lipid nanoparticle and surface-modified solid lipid nanoparticle delivery systems for oral delivery of phyto-bioactive compounds in various chronic diseases. Int J Nanomedicine, 2018. 13: p. 1569-1583.
Nooli, M., et al., Solid lipid nanoparticles as vesicles for oral delivery of olmesartan medoxomil: formulation, optimization and in vivo evaluation. Drug Dev Ind Pharm, 2017. 43(4): p. 611-617.
Shi, L.-L., et al., Positively Charged Surface-Modified Solid Lipid Nanoparticles Promote the Intestinal Transport of Docetaxel through Multifunctional Mechanisms in Rats. Molecular Pharmaceutics, 2016. 13(8): p. 2667-2676.
7. Line 66 changed to “has attracted”
8. Line 180 and 203 have changed ‘;’ with ‘:’
9. Line 230 had changed to [AUC]A instead of [AUC]A and [AUC]B of [AUC]B.
Round 2
Reviewer 1 Report
the manuscript was revised according to the reviewer's comments.